# Naturally Occurring Mutations within HBV Surface Promoter II Sequences Affect Transcription Activity, HBsAg and HBV DNA Levels in HBeAg-Positive Chronic Hepatitis B Patients

**DOI:** 10.3390/v11010078

**Published:** 2019-01-18

**Authors:** Ran Hao, Kuanhui Xiang, Yan Shi, Dong Zhao, Huifang Tian, Baohong Xu, Yufang Zhu, Huan Dong, Hai Ding, Hui Zhuang, Jie Hu, Tong Li

**Affiliations:** 1School of Nursing, Hebei Medical University, Shijiazhuang 050000, Hebei, China; haoran810720@sina.com (R.H.); yufangzhu2018@163.com (Y.Z.); donghuan132@sina.com (H.D.); 2Department of Microbiology and Infectious Disease Center, School of Basic Medical Sciences, Peking University Health Science Center, Beijing 100191, China; xiangkuanhui174@126.com (K.X.); zhuangbmu@126.com (H.Z.); 3Institute of Microbiology, Shijiazhuang Center for Disease Control and Prevention, Shijiazhuang 050000, Hebei, China; hbsy80@126.com (Y.S.); aileenzdongdong@163.com (D.Z.); fang45530@126.com (H.T.); wsws1120@126.com (B.X.); 4Hunan Sansure Biotech Incorporation, Lusong Road, Changsha 410000, Hunan, China; haid@sansure.com.cn

**Keywords:** hepatitis B virus, surface promoter, HBsAg, mutation, HBeAg-positive, C genotype

## Abstract

Mutations in hepatitis B virus (HBV) surface promoter II (SPII) have not been well studied in hepatitis B e antigen (HBeAg)-positive chronic hepatitis B (CHB) patients. We aimed to investigate SPII mutations in such patients and their biological and clinical impacts. Direct sequencing was used to detect SPII mutations in 106 HBeAg-positive treatment-naïve CHB patients with genotype C (82.1% (87/106) was C2) HBV infection. Results showed that mutation frequency in transcription factor (TF) unbinding region was significantly higher than that in TF binding region of SPII (C1: 3.4% vs. 1.3%; C2: 2.6% vs. 1.3%; *p* < 0.0001). Luciferase assay revealed distinct promoter activities among SPII mutants; especially SPII of G120A mutant had a 15-fold higher activity than that of wild-type (*p* < 0.001). In vitro experiments in HepG2 cells showed that G82A, A115C and G120A mutants increased the hepatitis B surface antigen (HBsAg) levels, while C18T had an opposite effect. G82A, A115C and G120A mutants boosted the intracellular HBV total RNA level. G120A mutation resulted in an increased HBV DNA level in vitro, consistent with the serological results in patients. Thus, novel SPII mutations would affect promoter activity, HBsAg, HBV DNA and HBV total RNA levels, suggesting their potential biological and clinical significances.

## 1. Introduction

Hepatitis B virus (HBV) is a common infectious agent in humans and causes more than 257 million chronic infections with increased risk of liver cirrhosis and hepatocellular carcinoma [1]. HBV is a partially double-stranded DNA virus and replicates via an error-prone reverse transcription step mediated by a reverse transcriptase (RT) [2,3]. Thus, its genome sequence exhibits distinct genetic variability [4]. The HBV genome consists of four overlapped open reading frames (C, P, S, and X), the transcription of which is controlled by four promoters (C, preS1, S, and X), respectively. Of note, surface promoter II (SPII) is responsible for preS2/S mRNA transcription, which directs the translation of middle and small hepatitis B surface proteins (MHBs and SHBs) [5].

Hepatitis B surface antigen (HBsAg) represents a valuable diagnostic seromarker of underlying HBV infection, and its level is widely adapted to clinical practice [6]. For pegylated interferon-treated patients, HBsAg level could be served as a stopping rule, and for nucleos(t)ide analogue-treated patients, a sharp reduction of HBsAg could be a predictor of HBsAg clearance in long-term treatment [7,8,9]. Remarkably, SHBs is the main constituent of HBsAg and regulated by SPII [10,11]. Thus, SPII is thought to play a central role in regulating HBsAg expression and serum level. Previous studies [6,7,8] indicated that treatment-naïve hepatitis B e antigen (HBeAg)-positive chronic hepatitis B (CHB) patients showed large fluctuations of serum HBsAg levels. Su et al. [9] found that some naturally occurring amino acid (AA) substitutions in HBV RT/S proteins might affect the serum HBsAg level in HBeAg-positive CHB patients. However, whether the mutations in SPII affect the HBsAg level have not been well characterized, and need further study.

The SPII promoter sequence is rich in guanine (G) and cytosine (C) nucleotides, and contains the CCAAT box and transcription initiation site. The transcription factors (TF) of nuclear factor (NF)-γ and CCAAT/enhancer binding protein (C/EBP) bind to the CCAAT box and are necessary for the SPII activity. The SPII could be divided into seven functional regions (A–G), which interact with sequence-specific DNA binding proteins to regulate transcriptions [12,13]. A, B, and C are positive regulatory regions, which could enhance SPII activity. The D region is homologous to the SV40 sequence and can be bound by a specific TF to regulate activity. The F region has a negative regulatory effect, but it can be countered by factors that bind to the E region. The G region contains a transcription initiation site and its deletion would abolish transcription.

Previous studies reported that core promoter (CP) was the hotspot region with mutations, impacting on viral replication at different stages of liver disease progression [14,15,16,17]. For instance, it was found that A1762T/G1764A were novel mutations within CP and could lead to decreased HBeAg expression but enhanced viral replication, which correlated with liver disease severity [15]. Nevertheless, only several investigations studied the affections of SPII mutations on HBsAg production or virion secretion. Sengupta et al. [18] reported that deletions in SPII were discovered in two patients with subgenotype A2 occult infection and associated with cytoplasmic aggregation of HBsAg compared to wild-type (WT) HBV in vitro study. Biswas et al. [19] reported that mutations in SPII disrupting the G-quadruplex in HBV genotype B could reduce its transcription activity, leading to reduced HBsAg levels and virion secretion.

The detailed analysis of mutations occurring within SPII in HBeAg-positive CHB patients prior to antiviral treatment is very limited, and its virological and clinical significance is not yet defined. In this study, SPII sequences in 106 treatment-naïve HBeAg-positive CHB patients with genotype C infection were analyzed to identify novel mutations potentially associated with serum HBsAg levels. Then, in vitro study was applied to verify the potential impacts of these novel mutations on promoter activity, HBsAg expressions and HBV replication.

## 2. Materials and Methods

### 2.1. Patients

HBV genotype C infected HBeAg-positive CHB patients without treatment at least six months prior to the recruitment were enrolled from a registered clinical study (NCT01088009) [20]. Informed consents were obtained. CHB diagnosis was according to the guidelines of prevention and treatment for CHB in China [21]. Exclusion criteria included other forms of liver diseases or human immunodeficiency virus co-infection, autoimmune liver disease, or alcohol or drug abuse. To avoid the impacts of AA substitutions in SHBs on serum HBsAg levels, the samples with any of 18 important AA substitutions in SHBs that affected HBsAg levels were excluded [22,23]. These AA sites were sE2, sL21, sR24, sT47, sI68, sC69*, sC76, sL95, sL98, sS117, sR122, sI126, sG145R, sV177, sW182*, sM198, sI218, and sV224. The screening process for patients was illustrated in Appendix A. The resulting cohort of 106 patients was studied.

### 2.2. Laboratory Tests for Patient Samples

HBsAg, anti-HBe and HBV DNA levels were measured as described in our previous studies [20]. The detection limit of plasma HBV DNA by the Roche TaqMan48 automatic florescence quantitative polymerase chain reaction (qPCR) kit was 12 IU/mL.

### 2.3. HBV DNA Amplification, Sequencing, and Sequence Analysis

HBV DNA was extracted from 200 μL serum samples using QIAamp DNA Blood Kit (Qiagen, Hilden, Germany). HBV SPII was amplified with a nested PCR method and the primer sequences were shown in Appendix A. Nested PCR products were subjected to direct sequencing of both directions (Sangon Bioengineering, Beijing, China).

The reference sequence of SPII was extracted from genotype C HBV genomic consensus sequence derived from our previous study [24]. SPII sequence was corresponding to nt2983–nt3210 in the genomic sequence and renumbered as nt1–nt228 in this study. Nucleotides mutations within SPII sequences were identified by comparing the individual sequence to the consensus sequence. All the samples had known HBV subgenotypes that were characterized based on SHBs sequences in our previous study [9].

### 2.4. Plasmids

The pGL3-basic reporter vector (Promega, Madison, WI, USA) was used for the constructions of SPII reporter plasmids according to the manufacture’s introduction and our previous study [25]. We constructed 13 such plasmids, i.e., WT, C18T, C44T, A75T, G82A, A115C, T118C, G120A, G129A, T134C, A138G, C189A, and G228A. The primers were listed in Appendix A. The WT SPII clone (pGL3-SPII) was made first, and then mutant clones were constructed by a site-directed mutagenesis strategy based on pGL3-SPII.

To assess the expression of HBsAg affected by mutant SPII, mutation of C18T, G82A, A115C, G120A, A138G, and C189A within SPII was also introduced into an HBsAg expressing plasmid of pBluescript-LMS (pLMS), respectively [22]. Primers for site-directed mutagenesis were shown in Appendix A.

Furthermore, mutants C18T, G82A, A115C, G120A, A138G, and C189A in SPII were cloned into a plasmid pBlueBac4.5 1.2/PC (p1.2/PC) respectively, which contained a 1.2-fold length HBV genome of genotype C2 with a G1896A mutation in the preC region [22]. Primers for homologous recombination were shown in Appendix A.

### 2.5. Cell Cultures, Transfection and Luciferase Assay

HepG2 cells were maintained in Dulbecco’s modified Eagles medium (DMEM) with 10% fetal bovine serum (Gibco, New York, NY, USA). Cells were seeded 24 hours before transfection with approximately 70% confluence, and then cells were transfected with HBV SPII WT or mutant recombinant plasmids (pLMS or p1.2/PC). The ratio of plasmids and transfection reagent of X-tremeGENE 9 (Roche diagnostics, Mannheim, Germany) was 1 μg:3 μL. After overnight incubation, the cells were washed with PBS three times and fresh media was added [22]. Transfected cells and supernatant were individually harvested 72 hours post-transfection.

A dual-luciferase reporter assay system (Promega, Madison, WI, USA) was used to test the promoter activities of pGL3-SPII recombinants. Any of the mutant pGL3-SPII plasmids with a pRL-TK vector (Promega; Renilla luciferase expression vector as an internal control) were co-transfected into cells. After overnight incubation, the cells were lysed, and luminescence was detected by an EnSpire Multimode reader (PerkinElmer, Norwalk, CT, USA).

### 2.6. Detection of HBsAg, HBV DNA, HBV Total RNAs and Pregenomic RNA in Cell Culture System

HBsAg from HepG2 cell lysate and supernatant was detected with a commercial enzyme-linked immunosorbent assay (ELISA) kit (Autobio, Zhengzhou, China). HBV DNA from cell lysate and supernatant was measured by a quantitative fluorescent diagnostic kit (Sansure Biotech, Changsha, China). The tests were all carried out according to the manufacturer’s protocols.

Intracellular total RNA was extracted with the RNeasy Mini Kit (Qiagen, Hilden, Germany), and quantified by NanoDrop (Thermo Scientific, Waltham, MA, USA). We performed reverse transcription-qPCR (RT-qPCR) assay to detect HBV total RNA and pregenomic RNA (pgRNA). The reverse transcription was carried out using EasyScript One-Step RT-PCR SuperMix (TransGen Biotech, Beijing, China). The primer sequences for RT-qPCR were shown in Appendix A. Ribosomal protein S11 mRNA was used to normalize RT-qPCR results for HBV transcripts.

### 2.7. Statistical Analyses

SPSS 21.0 (SPSS Inc., Chicago, IL, USA) was used for statistical analysis. Student’s *t*-test or Mann–Whitney test was used to examine the difference for continuous data where it was appropriated. Chi-squared test or Fisher’s exact test was applied for categorical data. All *p* values were two-tailed. A *p* < 0.05 was considered to be statistically significant.

### 2.8. GenBank Accession Numbers

The HBV SPII sequences obtained in this study were deposited in GenBank with accession numbers of MK263360–MK263465.

## 3. Results

### 3.1. Patient Characteristics

One hundred and six treatment-naïve HBeAg-positive patients with HBV C-genotype infection were qualified for this study. Of 106 patients, 82.1% (87/106), 17.0% (18/106) and 0.9% (1/106) were infected with the C2-subgenotype, C1-subgenotype and C5-subgenotype, respectively. The demographic and virologic characteristics of the patients were summarized in Table 1. A total of 75.5% of the patients was male. The median of HBV DNA level was 8.1 log IU/mL (6.6–9.8 log IU/mL) and the median HBsAg level was 4.4 log IU/mL (3.1–5.2 log IU/mL).

### 3.2. Comparison of SPII Mutations in C1 and C2 Sequences

In the present study, C1 and C2 subgenotypes accounted for 99.1% (105/106). In this section, we compared the patients and SPII sequences in C1/C2 subgenotype cohorts. Firstly, the demographic and virological characteristics of C1/C2-subgenotype cohorts were compared. As shown in Appendix A, the age in C1 group was significantly younger than that in C2 group (*p* = 0.0003), while the gender ratio, serum HBV DNA, and HBsAg levels in two groups were comparable.

Secondly, C1/C2 SPII mutations at individual nucleotide site were determined by comparing with the subgenotype specific consensus sequence, respectively. About 15.8% (36/228) nucleotide sites in C1 sequences and 37.7% (86/228) in C2 sequences were found with mutations, suggesting more variable sites detected in C2 sequences (*p* < 0.0001). The total mutation frequencies showed a slight difference between C1 group (2.5%, 103/(228 × 18)) and C2 group (2.0%, 405/(228 × 87)) (*p* = 0.0578) (Figure 1A). Furthermore, as shown in Figure 1A, the mutation frequencies of TF unbinding region in C1 sequences (3.4%, 85/(140 × 18)) was significantly higher than that in C2 sequences (2.6%, 316/(140 × 87)) (*p* = 0.0298). However, there was no significant difference for the mutation frequencies in TF binding regions (88 bp) between C1 and C2 sequences. We also found that the nucleotide sites in TF unbinding region were more prone to mutate than those in binding region, regardless of subgenotypes (*p* < 0.0001) (Figure 1B).

Thirdly, we compared the mutations in the various structural regions of SPII in C1 and C2 sequences. The results shown in Appendix A indicated that there was no statistical difference of the mutation frequency in each region of C1 and C2 sequences, respectively. Of note, the region C had the highest mutation frequencies in both subgenotypes (C1: 4.7%, 31/(37 × 18); C2: 4.1%, 132/(37 × 87)). Figure 2 showed the mutation frequency at each nucleotide site of SPII. The high-frequency mutation sites were similar for C1 and C2 SPII. They appeared in the 5’-end of NF1 binding region, the middle of region A, the inter-region of A and B and the 3’-end of region C. A peak of the mutations occurred at the 3’ end of the C region and that was a T134C mutation.

### 3.3. Mutation Analysis on the C2 SPII Sequences

Since 82.1% of the patients (*n* = 87) were infected by subgenotype C2 HBV, we then focused on the mutation analysis on the C2 SPII sequences. To study the impact of SPII mutations on HBsAg levels, 87 patients were divided into a low HBsAg group (LS, *n* = 43) and a high HBsAg group (HS, *n* = 44) according to median of HBsAg levels (23,648.0 IU/mL). As shown in Appendix A, there was no difference in age and gender ratio between the two groups. However, the HBV DNA level in the HS group was significantly higher than that in the LS group. In addition, the total mutation frequency within SPII in the LS group was significantly higher than that in the HS group (2.2%, 220/(228 × 43) vs. 1.8%, 185 /(228 × 44), *p* = 0.0465). The mutations were analyzed for different regions of SPII (Appendix A). The only significant difference in mutation frequencies between groups occurred in region A of SPII (LH vs. HS: 3.0%, 80/(63 × 43) vs. 1.9%, 54/(63 × 44); *p* = 0.018). Furthermore, 29.8% (68/228) and 27.2% (62/228) of nucleotide sites of SPII sequences in LS and HS groups were found to have mutations (*p* = 0.7540), in which 41 sites were shared by two groups (Figure 3). The difference in values of the mutation frequency at each mutation site, between LS/HS groups, was plotted in Appendix A. The results showed that the LS group had higher mutation frequencies at 46 sites compared to those in the HS group. As shown in Figure 3 and Appendix A, the distribution of the hottest mutation sites in two groups seemed not to be co-incident, such as C18T, G82A, G120A, and A138G. These high frequent mutations and some other potentially important mutations were worthwhile for further investigation in vitro.

### 3.4. The Influence of SPII Mutations on Promoter Activity

For in vitro study, we selected the following mutations based on criteria as follows. (1) Those sites with high mutation frequencies might have potential clinical significance. A high mutation site was defined when its mutation frequency was greater than 10% (five-fold higher than the overall mutation frequency of SPII sequences). (2) Those sites with mutation frequency difference greater than 3.0% between the LS and HS groups. Accordingly, 12 potential mutations were selected, i.e., C18T, C44T, A75T, G82A, A115C, T118C, G120A, G129A, T134C, A138G, C189A, G228A.

Twelve mutants in pGL3-SPII recombinants were analyzed in a dual luciferase reporter gene assay system. As shown in Figure 4, each mutation had a significant effect on the transcription activity of SPII (*p* < 0.01 or < 0.001). In comparison with WT, most of the mutants showed enhanced transcription activity, only C18T could inhibit the activity of SPII. The SPII activity of G120A mutant was 15-fold higher than WT.

Furthermore, the frequency of G120A mutation was 18.2% in the HS group (8/44), significantly higher than that in the LS group 2.3% (1/43) (*p* = 0.0298). Then, we categorized 87 cases into two groups based on G120A detected or not. Interestingly, significant statistical differences in virologic characteristics were found between two groups. The HBsAg levels were higher in G120A group than that in G120 WT group ((4.7 ± 0.4) log IU/mL vs. (4.3 ± 0.5) log IU/mL, *p* = 0.0040), shown in Figure 5A. Similarly, HBV DNA levels were higher in the G120A group than that in the G120 WT group ((8.5 ± 0.7) log IU/mL and (8.1 ± 0.6) log IU/mL, *p* = 0.0237), shown in Figure 5B.

### 3.5. Influence of SPII Mutations on HBsAg Levels and HBV Replication

Based on the mutant SPII transcription activities, we selected six mutants to study the phenotype of SPII mutants in vitro. For up-regulation, the top five mutants (G82A, A115C, G120A, A138G and C189A) were selected and for down-regulation, mutant C18T was selected. To study whether mutations of SPII affected HBsAg level, we constructed the recombinant plasmid pLMS with mutations including C18T, G82A, A115C, G120A, A138G and C189A, respectively. After transfecting HepG2 cells, HBsAg levels were detected by ELISA. As shown in Figure 6A and B, G82A, A115C and G120A showed higher extracellular and intracellular HBsAg levels compared to WT (*p* < 0.05). In addition, C18T exhibited the down-regulating effect of HBsAg (*p* < 0.05). However, A138G and C189A mutants had similar activities as the WT.

To further study the effect of SPII mutations on HBV transcription and replication, the above mutations (C18T, G82A, A115C, G120A, A138G and C189A) were also constructed into the p1.2/PC plasmid containing 1.2-fold HBV genome. The results showed that several SPII mutations affected HBsAg levels. Specifically, mutations G82A, A115C and G120A also resulted in higher extracellular and intracellular HBsAg levels compared to WT (*p* < 0.05) (Figure 6C,D). The extracellular and intracellular HBsAg levels from mutant C18T were lower than those from WT (*p* < 0.05) (Figure 6C,D), consistent with the results of the luciferase assay.

Due to SPII overlapping with a spacer region in the polymerase gene, SPII mutation may potentially impact on HBV transcription and replication. Thus, we detected the levels of HBV DNA, pgRNA and HBV total RNA from these mutations in replication system with p1.2/PC mutant transfection. The SPII mutants had no effect on HBV pgRNA levels, as shown in Figure 7A. The results showed that HBV total RNA levels from G82A, A115C and G120A were significantly higher than that of WT (*p* < 0.05) (Figure 7B), while C18T down-regulated that (*p* < 0.05) (Figure 7B). We also analyzed the effect of the six mutants on extracellular and intracellular HBV DNA and found that G120A up-regulated both the extracellular and intracellular DNA levels (Figure 7C,D). Overall, these results suggest that mutations in SPII may regulate the HBV mRNA transcription levels and resulted in the changed expression levels of HBsAg.

## 4. Discussion

In this study, mutations within HBV SPII region in the C-genotype infected treatment-naïve HBeAg-positive CHB patients were specifically investigated. We identified that naturally occurred mutations within SPII were prevalent, which have not been well studied yet. By analyzing the incidence of mutations in the SPII region and the virological characteristics in patients, it was presumed that some mutations might regulate HBsAg levels. Further functional study on some novel mutations in SPII region, such as G82A, A115C, G120A and A138G and so on, revealed that these mutations could affect the transcription activity of SPII promoter regulating HBsAg levels and HBV replication.

We successfully obtained 106 HBV SPII sequences from 113 enrolled patients. There were seven samples with failed sequencing, due to the presence of multiple peaks on the sequencing map. Further clone sequencing revealed that some deletions and insertions existed in SPII, which were consistent with the previous study [18,24]. Of the 106 patients with successful SPII sequencing, 18 and 87 patients were infected with subgenotypes C1 and C2 HBV, respectively. It did not show any significant differences of HBsAg levels between these two subgenotypes. Correspondingly, there was no significant difference of the total mutation frequency in the full-length SPII region between these two subgenotypes (C1 and C2: 2.5% and 2.0%, *p* = 0.0578). However, we speculated the *p* value suggesting that there might be some differences between the two subgenotypes. Further analysis showed that the differences in mutations in the SPII region of the C1 and C2 groups were mainly reflected in the TF unbinding regions. There was a significant difference in the mutation frequency in this region between these two groups (3.4% and 2.6%, *p* = 0.0298). In addition, regardless of the C1 or C2 subgenotype, the mutation frequency of the TF binding region was significantly lower than that of the TF unbinding region (Figure 1B). These results suggest that the TF binding regions are important functional domains with a relatively conserved sequence to ensure the normal transcription activity.

Interestingly, when 87 patients infected with C2-subgenotype were divided into LS and HS groups based on the median serum HBsAg level, we demonstrated that the total mutation frequency in the SPII in the LS group was significantly higher than that in the HS group (*p* = 0.0465, see Appendix A), indicating that mutations in this region may regulate viral protein expression. In region A, the detected mutation frequency in the LS group was significantly higher than that in the HS group (*p* = 0.0178). This indicates that the existence of SPII mutations might regulate its activity, resulting in a changed HBsAg expression. In addition, it was noteworthy that HBV DNA levels were also significantly lower in the LS group than that in the HS group. Therefore, the lower HBsAg levels in the LS group may also be associated with lower overall virus replication. As mentioned earlier, HBsAg levels are affected by a combination of factors. To verify the impact of SPII mutations on transcription activity and viral replication, we screened out valuable hotspot sites for in vitro functional study.

By luciferase assay in cell culture system, we demonstrated that only C18T mutation among the 12 SPII novel mutations significantly inhibited SPII activity. However, other mutations led to increased SPII activity, especially the G120A mutation showing 15-fold higher activity than WT. Further, we divided 87 C2-subgenotype cases into two groups according to the G120 mutation. Results showed that HBsAg and HBV DNA levels were significantly higher in the G120A mutation group than those of WT (*p* < 0.05). These results indicate that G120A mutation may impact viral replication. The G120 is located in the C region of SPII that plays a positive regulatory role in HBV transcription [10]. Although G120A was only a single mutation, our results suggest that it might up-regulate the transcriptional activity of SPII. Interestingly, the G120A mutation increased HBV DNA levels in vitro (see Figure 7), which was consistent with the patient results. Thus, its clinical relevance is a distinct possibility. Whether single-site mutations may have such a significant effect on activity, we refer to the effects of the A1762T/G1764A mutation in HBV [14,15,16,17]. Moreover, we found that most of these hotspot mutations were not prone to mutate in combinations (C18T + G82A (*n* = 1), C18T + A115C (*n* = 1), C18T + A138G (*n* = 1) and A115C + G120A (*n* = 1)), however, G82A + A138G combination mutations were detected in 17.2% (15/87) of C2 sequences. If the combination mutations have any synergistic effects, further studies are needed.

After verified by luciferase assay, novel SPII mutants were further constructed into vectors pLMS and p1.2/PC. Mutants A115C, G120A and A138G upregulated extracellular/intracellular HBsAg levels (Figure 6), and C18T downregulated those (Figure 6), which were consistent with luciferase assay (Figure 4). At the same time, C18T, A115C, G120A and A138G could regulate HBV total RNA levels, but they have no effect on pgRNA levels. There might be three possible reasons for these results. Firstly, it is speculated that A115C, G120A and A138G in the C region of SPII may enhance the promoter activity, since the C region is a positive regulatory region. However, C18T locates in TF unbinding region, the significant impact of this mutation hints that the inter-functional regions might have unrevealed functions. The second reason is that the SPII and PreS1 regions are overlapped. The A115C mutation in SPII corresponds to PreS1-I84L substitution, but G120A and A138G do not cause any AA changes of PreS1. Previous studies [26,27] found that aa57–90 of PreS1 belongs to the hydrophobic region, and was an important structural and functional domain. Mutations in this region may regulate the role of post-transcriptional translation by changing the hydrophobic force. Therefore, it is speculated that A115C mutation may affect HBsAg expression. Thirdly, due to the spacer region of the P gene overlapping with SPII, mutations A115C, G120A, and A138G of SPII correspond to spacer-H87P, D89N and T95N AA substitution, respectively. Studies [28,29] have shown that spacer is a highly mutated region that seems to be able to effectively balance the distinctive conservation and variation requirements occurring within the overlapping region with PreS1/SPII. Considering the overlap feature of the HBV genome, some mutations in SPII may cause concomitant AA substitutions in PreS1 or spacer region of HBV polymerase. Though, to study the mutation impacts on the PreS1 (LHBs) and HBV polymerase function seems to beyond the scope of this investigation, it deserves for further study.

Although C18T, G82A, A115C and G120A showed similar behaviors regarding regulation of HBsAg expression in three different in vitro systems, respectively, the up-regulation effect of A138G and C189A on HBsAg expression found in luciferase reporter system was not confirmed in pLMS and p1.2/PC systems. The first possible reason was that luciferase was a single protein but pLMS produced three HBsAg proteins (LHBs/MHBs/SHBs), among which LHBs was regulated by SPI, and MHBs/SHBs were regulated by SPII [3]. SPII has been reported to be also significantly activated (up to 10-fold) by increased amounts of LHBs [3]. That is to say, the roles of SPI and SPII may have interplays. In addition, the LHBs/MHBs/SHBs might also form subviral particles (SVP), and the secretion pathways of spherical and tubular SVP were different [30]. Therefore, there may be more influencing factors in pLMS and p1.2/PC systems that deserve further investigation.

There are three main limitations in our study. Firstly, we focused on HBeAg-positive CHB patients infected with C-genotype, especially with C2-subgenotype. It is better to do mutation analysis for other (sub)genotypes. Secondly, we just focused on the mutation distribution within SPII for the treatment-naïve patients. However, the dynamic changes of SPII mutations during antiviral treatment have not been investigated. Thirdly, the precise regulation of novel SPII mutations on MHBs/SHBs and the mechanism in vivo needs further in-depth exploration.

## 5. Conclusions

In summary, this study analyzed the mutations in HBV SPII region in a cohort of 106 HBeAg-positive CHB treatment-naïve patients with HBV C-genotype infection. It was found that SPII mutations were prevalent in TF unbinding regions. Finally, we identified several novel mutations, such as C18T, A115C, G120A and A138G, which could regulate the SPII promoter activity and HBsAg levels. Especially, in vitro studies suggest that SPII G120A might play a positive role in up-regulating the HBV total RNA to increase the HBsAg and HBV DNA levels. Its potential clinical significance in treatment-naïve patients requires more attention. To better evaluate the SPII mutations will meet the needs of individualized patient management.

## Figures and Tables

**Figure 1 viruses-11-00078-f001:**
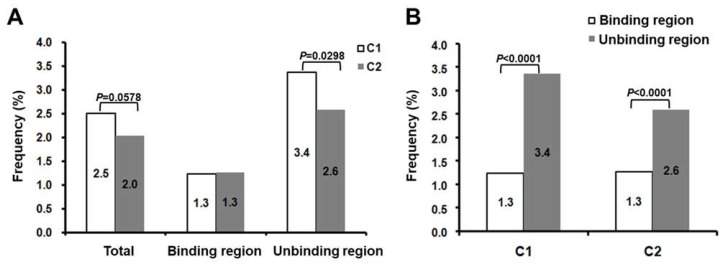
The comparison of mutations detected in C1 and C2 sequences. (**A**) Mutations occurred in the entire surface promoter II (SPII) region (228 bp), transcription factor (TF) binding region (88 bp) and TF unbinding region (140 bp). (**B**) The difference of mutation frequency between the TF binding region and the unbinding region was shown for the C1 and C2 subgenotypes. Chi-square tests were applied.

**Figure 2 viruses-11-00078-f002:**
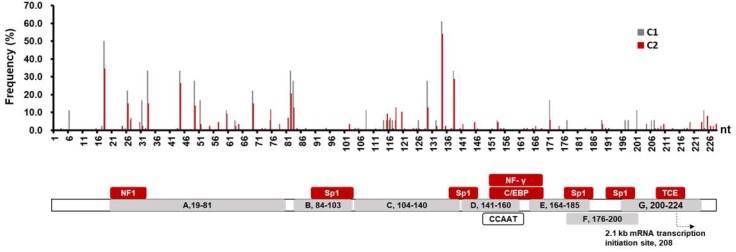
The mutation frequency at individual nucleotide sites within the SPII sequences in hepatitis B virus (HBV) C1 and C2 subgenotypes. The schematic diagram of HBV SPII is shown below the bar chart. The structural regions A to G are presented in grey boxes [10,11,12]. The regions that the transcription factors (TFs) bound are presented in red boxes. NF, nuclear factor; Sp1, specificity protein 1; C/EBP, CCAAT/enhancer binding protein; TCE, transcriptional control element.

**Figure 3 viruses-11-00078-f003:**
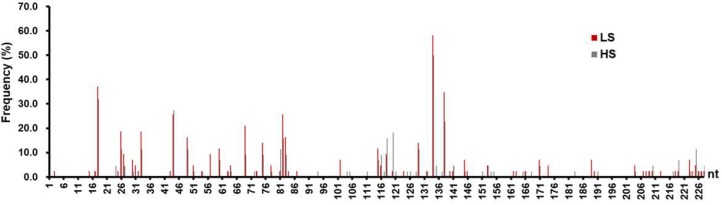
The mutation frequency at individual nucleotide sites within SPII sequences in low hepatitis B surface antigen (HBsAg) group (LS) and high HBsAg group (HS) patients infected with C2-subgenotype HBV. According to the median level (23,648.0 IU/mL), 87 patients with C2 infection were divided into 43 cases with low HBsAg (LS), and 44 cases with high HBsAg (HS). The grey/red bars indicated LS and HS groups, respectively.

**Figure 4 viruses-11-00078-f004:**
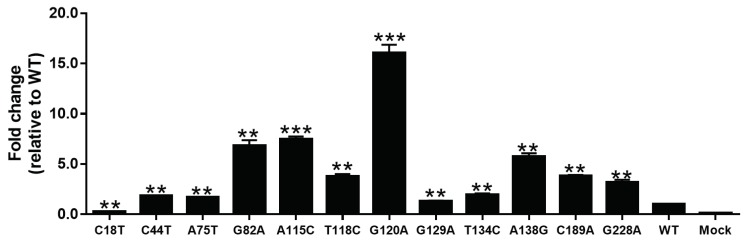
Fold change of the relative luciferase activities of SPII sequences containing different point mutations relative to that of the wild-type (WT) C2 SPII. Mann–Whitney U tests were employed. ** *p* < 0.01; *** *p* < 0.001.

**Figure 5 viruses-11-00078-f005:**
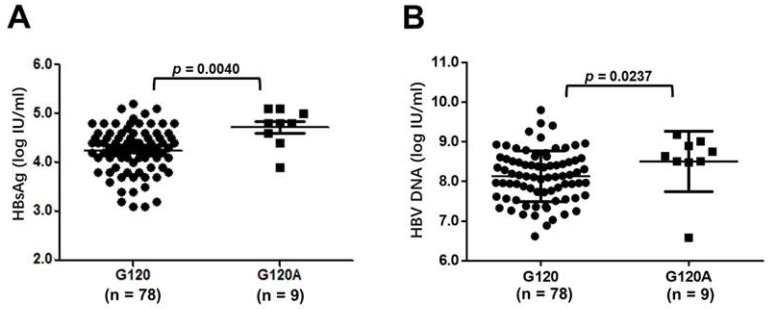
Scatter plots presenting levels of HBsAg (**A**)/HBV DNA (**B**) for 87 chronic hepatitis B (CHB) C2-subgenotype patients without/with G120A mutation. According to Mann–Whitney U tests, the virological characteristics between the two groups were statistically different (*p* = 0.0040 for HBsAg and *p* = 0.0237 for HBV DNA, respectively).

**Figure 6 viruses-11-00078-f006:**
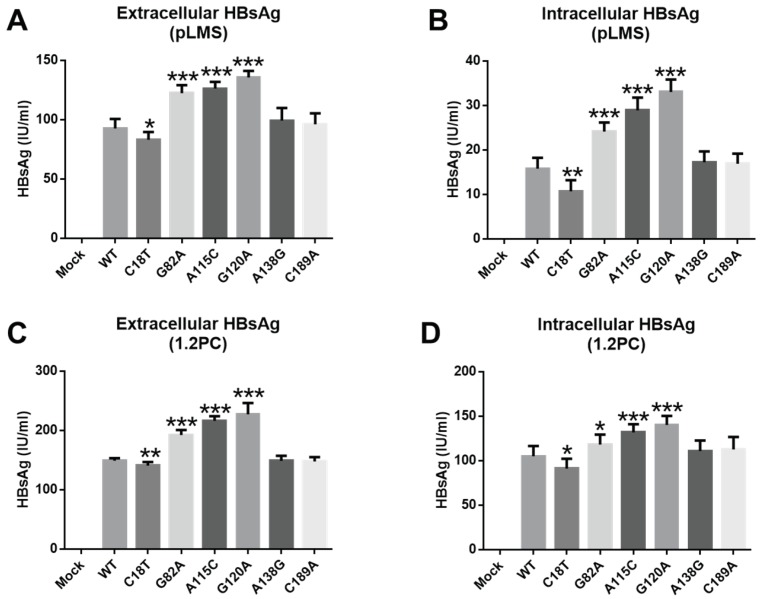
The impact of HBsAg expression regulated by the SPII mutants in HepG2 cells. Mutants were construct into the background of pLMS (**A**,**B**)/p1.2/PC (**C**,**D**), respectively. After 72 hours transfection into HepG2 cells, supernatants and cell lysate were harvested for detection of hepatitis B surface antigen (HBsAg) by ELISA. Mann–Whitney U tests were employed. * *p* < 0.05; ** *p* < 0.01; *** *p* < 0.001.

**Figure 7 viruses-11-00078-f007:**
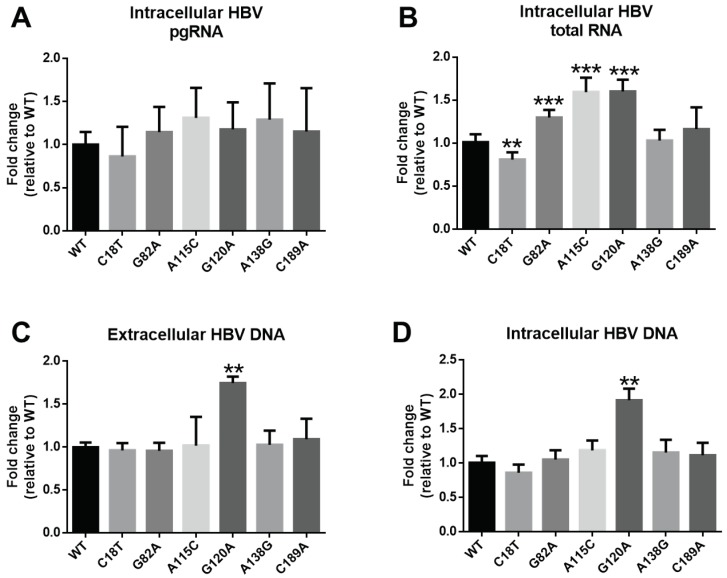
Comparison of HBV replication between wild-type (WT) and mutants SPII in HepG2 cells. The p1.2/PC-based SPII WT and mutant plasmids were transfected into HepG2 cells that were harvested 72 hours after transfection. Total RNA was isolated for detection of HBV total RNA (**A**) and pregenomic RNA (pgRNA) (**B**) using RT-qPCR. HBV DNA from cell supernatant (**C**) and cell lysate (**D**) was measured by qPCR. Data shown as fold change relative to WT. Mann–Whitney U tests were employed. ** *p* < 0.01; *** *p* < 0.001.

**Table 1 viruses-11-00078-t001:** Demographic and virologic characteristics of the studied patients.

Characteristics	Value (*n* = 106)
Age, median (range)	31.5 (18–58)
Gender M/F (% M)	80/26 (75.5)
HBV DNA (log IU/mL), median (range)	8.1 (6.6–9.8)
HBsAg (log IU/mL), median (range)	4.4 (3.1–5.2)

M, male; F, female; HBsAg, Hepatitis B surface antigen.

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
