# Peer review of "Naturally Occurring Mutations within HBV Surface Promoter II Sequences Affect Transcription Activity, HBsAg and HBV DNA Levels in HBeAg-Positive Chronic Hepatitis B Patients"

_viruses, 2019, doi:10.3390/v11010078_

Reviewer 1 Report

The manuscript by Hao et al describes naturally occurring mutations within the surface promoter II region in a treatment naïve HBe-Ag positive patient cohort. Mutations tended to spare known transcription factor binding sites. In an effort to assess the possible consequences of mutations for HBs-AG expression, the subgenotype C2 patients were divided into high vs. low HBS subgroups, which revealed an apparent correlation of higher HBV DNA levels with higher HBsAG levels. The p-value of 0.0000 would seem to be an error; the inclusion of a p-value for the HBsAg difference is meaningless, as the subgroups were defined by this value.

To further explore the significance of the mutations, several were chosen for cell based studies. First, the mutations were individually engineered into a luciferase reporter construct. Several mutations produced higher expression of luciferase, with one down-regulating the reporter. Figure 4 is somewhat confusing, as WT appears to be used as a reference, but the plot shows it with a value less than 1.0. This needs clarification. 

The mutation that had the strongest effect on promoter activity, G120A, also appeared to be more frequent in the high HBsAg subgroup. The same patient set was re-analyzed by the presence or absence of the mutation, indicating that the presence of the mutation correlated with both elevated HBsAg and HBV DNA levels. The authors attempted to further explore these correlations by engineering the mutations in an HBV genome construct. Differences in general were small, but in general HBsAg expression followed the luciferase results. Interestingly, the G120A mutation increased HBV DNA levels, consistent with the patient results. Thus, clinical relevance is a distinct possibility.

Other comments:

-DNA sequencing method used is not described.

-Are the Y-axis values in figures 2 and 3 correct? They seem inconsistent with the values in the text.

- Minor errors in English

Author Response

Response to Reviewer 1 Comments

1.    The manuscript by Hao et al describes naturally occurring mutations within the surface promoter II region in a treatment naïve HBeAg positive patient cohort. Mutations tended to spare known transcription factor binding sites. In an effort to assess the possible consequences of mutations for HBsAg expression, the subgenotype C2 patients were divided into high vs. low HBS subgroups, which revealed an apparent correlation of higher HBV DNA levels with higher HBsAg levels. The p-value of 0.0000 would seem to be an error; the inclusion of a p-value for the HBsAg difference is meaningless, as the subgroups were defined by this value.

Response 1: Thank you very much for your suggestion. We agree with you and have revised our supplementary Table S4 accordingly.

2.   To further explore the significance of the mutations, several were chosen for cell based studies. First, the mutations were individually engineered into a luciferase reporter construct. Several mutations produced higher expression of luciferase, with one down-regulating the reporter. Figure 4 is somewhat confusing, as WT appears to be used as a reference, but the plot shows it with a value less than 1.0. This needs clarification. 

Response 2: The Y-axis value represents the relative luciferase activity, that is the ratio of firefly (Photinus pyralis) and Renilla (Renilla reniformis) luciferase. According to your suggestion, we reprocess the data using the WT as a reference (please see the revised Figure 4). Thus, the value of WT is 1.0. Re-calculation does not affect the statistical differences.

3.    The mutation that had the strongest effect on promoter activity, G120A, also appeared to be more frequent in the high HBsAg subgroup. The same patient set was re-analyzed by the presence or absence of the mutation, indicating that the presence of the mutation correlated with both elevated HBsAg and HBV DNA levels. The authors attempted to further explore these correlations by engineering the mutations in an HBV genome construct. Differences in general were small, but in general HBsAg expression followed the luciferase results. Interestingly, the G120A mutation increased HBV DNA levels, consistent with the patient results. Thus, clinical relevance is a distinct possibility.

Response 3: We appreciated your summarized analysis on our results. We are very glad to include some of your suggestions in the Discussion section (please see the 4th paragraph in Discussion).

4.    DNA sequencing method used is not described.

Response 4: The detailed information has been added in the revised manuscript (please see 2.3 in Materials and Methods).

5.    Are the Y-axis values in figures 2 and 3 correct? They seem inconsistent with the values in the text.

Response 5: We are sorry for the confusion. There are several statements about the data shown in Figures 2 and 3. We clarify them one by one as follows. Firstly, the mutation frequency at individual nucleotide site (the Y-axis values in Figures 2 and 3), is calculated by the following formula: (mutation numbers at individual nucleotide sites of the analyzed SPII sequences/total numbers of the analyzed SPII sequences)´100%. For example, in Figures 2, the mutation numbers at the 18th nucleotide sites of C2 subgenotype is 30 and the total number of the C2 SPII sequences analyzed is 87, so the mutation frequency at the 18th site for C2 subgenotype is 34.5%, (30/87)´100%. Secondly, the total mutation frequency (mentioned in Results 3.3), is calculated by the following formula: [total mutation numbers at all nucleotide sites of the analyzed SPII sequences/(total numbers of the analyzed SPII sequences´total nucleotide numbers of SPII sequence)]´100%. For example, in the 1st paragraph in Results 3.3, the total mutation numbers in LS group is 220, the total numbers of the LS SPII sequences is 43 and the total nucleotide numbers of SPII sequence is 228, so the total mutation frequency for LS group is 2.2% [220/(228´43) ´100%]. To avoid this confusion, we have revised the title of Figure 2 and Figure 3.

6.    Minor errors in English.

Response 6: We have tried our best to revise our manuscript.

Reviewer 2 Report

Dr. Hao and colleagues presented an interesting study in which they aimed to explore the impact of naturally occurring mutations in the HBV SPII promoter on HBsAg and HBV DNA expression levels in CHB patients from China. The colleagues enrolled 106 CHB patients, determined the genotype and analysed the sequences of the SPII promoter by direct sequencing. Furthermore, the authors investigated identified mutations in functional cell culture experiments. As a result the authors identified and characterized 12 distinct mutations which may influence HBV replication and HBsAg production. The authors concluded from their findings that the novel SPII mutations would affect promoter activity and at the end HBV replication suggesting potential biological and clinical significance.

Overall, the presented study is well performed showing convincing data and yet concise in its content. The tables, figures and references are adequate. However, there are some comments which should be addressed.

Comments

1.    The English is appropriate; however, should be checked carefully at places.

2.    Material and Methods, line 87 and 90. The colleagues wrote that the patients received no treatment; however, this is in my opinion only true for six month before analyses?

3.    Material and Methods, line 90. What is AA substitution?  I guess it is amino acids but the abbreviation was not given.

4.    Material and Methods, line 92 ff. Why excluding a number of mutations in the SPII promoter sequence overlapping with SHBs? For sure, the consideration of the authors is correct that these specific mutations will interfere with HBsAg production due to other regulation than by promoter activity; it will be also of interest.

5.    Material and Methods, line 88. Was HDV co-infection excluded?

6.    The authors characterized 12 selected in luciferase assays for transcription activity and from these 6 distinct mutations for HBV replication in a replicon system using cell culture experiments. It is interesting to follow the results of these single mutations. However, it would be also of interest what kind of individual mutation pattern could be detected and which impact these combined mutations will have on HBV replication. In other words, single mutations and multiple mutations (as mainly present in natural HBV strains) can have compensatory effects. This aspect should be discussed.

7.    Did the authors perform rescue experiments? For example the mutation C18T is able to down-regulate HBsAg. Using cell culture experiments HBsAg production of the C18T-mutant plasmid could be compensated by wt-plasmid.

Are there differences in the mutation pattern or different mutations of the C1 and C2 strains?

Author Response

Response to Reviewer 2 Comments

1.    The English is appropriate; however, should be checked carefully at places.

Response 1: Thank you very much for your comments and suggestions. We have polished our English.

2.    Material and Methods, line 87 and 90. The colleagues wrote that the patients received no treatment; however, this is in my opinion only true for six months before analyses?

Response 2: Yes, you are right. We have revised our manuscript accordingly (see 2.1 in Materials and Methods).

3.    Material and Methods, line 90. What is AA substitution? I guess it is amino acids but the abbreviation was not given.

Response 3: This abbreviation has been given in line 54, where it appeared for the 1st time in the manuscript.

4.    Material and Methods, line 92. Why excluding a number of mutations in the SPII promoter sequence overlapping with SHBs? For sure, the consideration of the authors is correct that these specific mutations will interfere with HBsAg production due to other regulation than by promoter activity; it will be also of interest.

Response 4: SHBs (small hepatitis B surface proteins) is regulated by SPII. SPII sequence is corresponding to nt2983-nt3210 in the genomic sequence and SHBs coding region is corresponding to nt155-nt835. They don’t overlap with each other. To avoid the impacts of some well-known amino acid (AA) substitutions in SHBs on HBsAg expression, the sequences with any of the 18 important AA substitution(s) in SHBs that affected HBsAg levels were excluded. Thus, we can study the affection of nucleotide mutations in SPII sequences on HBsAg expression with no worry of AA substitutions in SHBs.

5.    Material and Methods, line 88. Was HDV co-infection excluded?

Response 5: Our study used some serum samples from a registered clinical study (NCT01088009), in which the enrolled patients had no other forms of liver diseases other than chronic hepatitis B. We have revised our manuscript accordingly (see line 87 in Materials and Methods).

 6.    The authors characterized 12 selected in luciferase assays for transcription activity and from these 6 distinct mutations for HBV replication in a replicon system using cell culture experiments. It is interesting to follow the results of these single mutations. However, it would be also of interest what kind of individual mutation pattern could be detected and which impact these combined mutations will have on HBV replication. In other words, single mutations and multiple mutations (as mainly present in natural HBV strains) can have compensatory effects. This aspect should be discussed.

Response 6: Thank you for your suggestion. We re-analyzed our data and found some combined mutations in C2 subgenotype (n=87), i.e., C18T+G82A (n=1), C18T+A115C (n=1), C18T+A138G (n=1), G82A+A138G (n=15, 17.2%), A115C+G120A (n=1). We have updated our manuscript accordingly (see the 4rd paragraph in Discussion).

7.    Did the authors perform rescue experiments? For example the mutation C18T is able to down-regulate HBsAg. Using cell culture experiments HBsAg production of the C18T-mutant plasmid could be compensated by wt-plasmid.

Response 7: In this study, we investigate the nucleotide mutations located in SPII, which is a regulatory region responsible for preS2/S mRNA transcription. As we know, the rescue experiments are usually used to study amino acid substitutions in a protein sequence, where the WT-plasmid could provide the wild-type protein to compensate the impaired function of the mutant. However, considering the overlap feature of HBV genome, some mutations in SPII may cause concomitant amino acid substitutions in PreS1 or spacer region of HBV polymerase. To study the mutation impacts on the PreS1 (LHBs) and HBV polymerase function seems to beyond the scope of this investigation, but it deserves for further study. We have discussed this issue (please see updated discussion in the fifth paragraph in Discussion).

 8.    Are there differences in the mutation pattern or different mutations of the C1 and C2 strains?

Response 8: We compared the mutations in the various structural regions of SPII in C1 and C2 sequences in Table S3. Figure 2 showed comparison of the mutation frequency at each nucleotide site of SPII between C1 and C2 subgenotypes. The high-frequency mutation sites were similar for C1 and C2 SPII. We compared the mutation frequency site by site and there was no statistical difference for each nucleotide site. In addition, we also compared the serum HBsAg levels between C1-infected patients with and without C18T, or G82A, or A115C, or G120A, or A138G or C189A mutations, but no statistical difference was found. This might be due to the small sample size of C1-infected patients or an overall difference of SPII in C1 and C2. Your suggestion is certainly worthwhile for a further study, in which more C1-infected patients will be recruited. Thank you very much.
